# Evidence-Based Surgery: What Can Intra-Operative Images Contribute?

**DOI:** 10.3390/jcm12216809

**Published:** 2023-10-27

**Authors:** Pietro Regazzoni, Jesse B. Jupiter, Wen-Chih Liu, Alberto A. Fernández dell’Oca

**Affiliations:** 1Department of Trauma Surgery, University Hospital Basel, 4031 Basel, Switzerland; 2Hand and Arm Center, Department of Orthopedics, Massachusetts General Hospital, Boston, MA 02114, USA; jjupiter1@mgb.org; 3Department of Orthopedics, Kaohsiung Medical University Hospital, Kaohsiung 80756, Taiwan; 4School of Medicine, College of Medicine, Kaohsiung Medical University, Kaohsiung 80756, Taiwan; 5Department of Traumatology, Hospital Britanico, Montevideo 11600, Uruguay; albertofernandez.uy@gmail.com; 6Residency Program in Traumatology and Orthopedics, University of Montevideo, Montevideo 11600, Uruguay

**Keywords:** randomized control trials, meta-analysis, guideline, evidence-based medicine, Evidence-Based Surgery, intra-operative technical performance quality, skills, homogeneity of data, reproducibility of research, scientific publication formats, learning tools, surgical decision making, shared decision making, patient-related outcome measurements, artificial intelligence and machine learning, patient autonomy

## Abstract

Evidence-based medicine integrates results from randomized controlled trials (RCTs) and meta-analyses, combining the best external evidence with individual clinical expertise and patients’ preferences. However, RCTs of surgery differ from those of medicine in that surgical performance is often assumed to be consistent. Yet, evaluating whether each surgery is performed to the same standard is quite challenging. As a primary issue, the novelty of this review is to emphasize—with a focus on orthopedic trauma—the advantage of having complete intra-operative image documentation, allowing the direct evaluation of the quality of the intra-operative technical performance. The absence of complete intra-operative image documentation leads to the inhomogeneity of case series, yielding inconsistent results due to the impossibility of a secondary analysis. Thus, comparisons and the reproduction of studies are difficult. Access to complete intra-operative image data in surgical RCTs allows not only secondary analysis but also comparisons with similar cases. Such complete data can be included in electronic papers. Offering these data to peers—in an accessible link—when presenting papers facilitates the selection process and improves publications for readers. Additionally, having access to the full set of image data for all presented cases serves as a rich resource for learning. It enables the reader to sift through the information and pinpoint the details that are most relevant to their individual needs, allowing them to potentially incorporate this knowledge into daily practice. A broad use of the concept of complete intra-operative image documentation is pivotal for bridging the gap between clinical research findings and real-world applications. Enhancing the quality of surgical RCTs would facilitate the equalization of evidence acquisition in both internal medicine and surgery. Joint effort by surgeons, scientific societies, publishers, and healthcare authorities is needed to support the ideas, implement economic requirements, and overcome the mental obstacles to its realization.

## 1. Introduction and Focus of Present Article

A review of a complex and disputed topic like Evidence-Based Surgery should include a detailed historical overview and a critical analysis of the most important available studies, synthesizing the resulting evidence in a usable form for a broad spectrum of readers [1,2]. Narrowing the focus of Evidence-Based Surgery to orthopedic surgery, rather than encompassing the entire field of surgery, necessitates the involvement of many subspecialists to highlight the specific issues pertinent to their respective fields. Such a project is unsuitable for a single article; a whole book is needed, like the one edited by Thoma et al. a few years ago [3]. The arguments discussed in the present article are derived from experiences with orthopedic trauma cases and focus on the details of intra-operative technical performance. It is legitimate to ask whether such experiences could be applied to other surgical domains. As surgical procedures are skill-dependent across the whole spectrum of surgical specialties, such an extrapolation seems acceptable. It would be desirable for similar documentation concepts to be expanded to other surgical fields. Specialties using endoscopy have some experience with complete image documentation, but the problems with handling huge data have—so far—limited broad secondary analysis.

The core principles of evidence-based medicine are established through many randomized controlled trials (RCTs) and meta-analyses. However, heterogeneity in surgical trials, such as surgeon-related factors (including surgeon preferences, factors (such as reluctance to participate and patient preferences), and methodological challenges (inclusion/exclusion criteria, sample size estimation, follow-up/drop-out rate, and funding can make it difficult to conduct RCTs in surgery [4,5,6]. Under certain circumstances, observational cohort studies may provide insights that are comparable to those of RCTs [7].

RCTs and systematic reviews provide valuable contributions to the search for the best available evidence. A compilation of different articles from different subspecialties can pinpoint specific problems but might not focus enough on common problems, such as the need for complete intra-operative image documentation [8]. Such data are a prerequisite for an essential element of scientific publications, the comparison of the technical performance quality and the analysis of the homogeneity of case groups. The homogeneity of case groups is essential for the reproducibility of results [6,9]. Sound decisions must rely on reproducible real-world data [10].

This paper, therefore, starts from the actual situation of evidence-based orthopedic surgery in a broad sense and focuses on the mentioned importance of intra-operative image data and the homogeneity of case groups. With our experience, involving a dataset of more than 1000 orthopedic trauma cases with complete peri-operative image documentation, including radiologic imaging, intra-operative surgical images, and corresponding follow-up images [11], we think we have valid arguments for the usefulness of such data. Orthopedic trauma is only a small segment of orthopedic surgery, but we feel comfortable drawing more extensive conclusions. For example, in a secondary workup of a surgical database containing comprehensive intra-operative imagery, we can analyze prevalent fractures such as the joint depression type of tibial plateau fractures [12]. Furthermore, we can also evaluate less common fractures like dorsal radiocarpal dislocations [13]. In this paper, we focus on the importance of intra-operative image data and do not expand on statistical, epidemiological, and outcome aspects. The importance of corresponding aspects is well-defined in the literature [10,14].

## 2. Evidence from Medical and Surgical Publications

### 2.1. Randomized Controlled Trials in Surgery

David Sackett and his colleagues defined Evidence-Based Medicine as “the conscientious, explicit and judicious use of current best evidence in making decisions about treating individual patients” [15].

RCTs, established and standardized by Austin Bradford Hill in the 1940s [16], are still considered the “gold standard” for clinical trials in many medical fields. Meta-analyses based on analyzing these RCTs systematic reviews are at the top of the evidence-based medicine pyramid [17]. A golden standard represents the best available data against which to test new data to evaluate the efficacy of a treatment [18].

What is the situation for surgical publications? The quality of intervention reporting in orthopedic trials, as showcased in published studies, is currently subpar. Anderson and his team have called for enhanced quality and comprehensiveness in these reports. They argued, drawing from studies utilizing the Consolidated Standards of Reporting Trials (CONSORT), that there is significant room for improvement in the methodological reporting of orthopedic trials [19]. Regarding the reasons, insufficient blinding, allocations, and other methodological deficits are listed. The Template for Intervention Description and Replication (TIDieR) was created to allow improvements [20], but it was not successful [21].

As the intervention-reporting quality determines the feasibility of reproducing a study, this situation needs urgent changes. Intervention modifications and intervention fidelity are the two items that are most often omitted. This gives rise to the potential for type I errors, where a non-significant outcome is incorrectly perceived as significant, and type II errors, where a significant outcome is mistakenly deemed non-significant. Modifications, like a change in the material used, must be documented. Intervention fidelity is equally important and should be analyzed. A failure to document such changes inevitably produces unacceptable inhomogeneity among the case groups, causing the whole set of results to become questionable.

The difficulty in creating homogeneous groups when developing surgical randomized controlled trials has been identified as an important problem [22]. In medical and surgical trials, it is possible to assign patients to different treatment arms. However, the evaluation of the medical outcome of the administration of a pharmaceutical may prove clearer than that of a surgical procedure [23]. One important reason could be the fact that contemporary documentation formats rarely allow the direct assessment of differences in surgical skills, which have direct impacts on the outcomes [24,25]. A visual representation conveys complex data more effectively than a textual description in surgical documentation [26]. Moreover, complete intra-operative image documentation allows us to analyze the skill-dependent differences in the technical performance [22].

Among the specific problems, the single factor of the skill-dependent quality of the intra-operative technical performance and its variations is not discussed seriously enough in the literature regarding Evidence-Based Surgery, despite its great importance regarding the outcomes of surgical treatment [27,28]. Skill measurements are often discussed when surgical training is concerned [29]. Still, skill variations among certified surgeons exist and are rarely directly analyzed and thematized with their outcome consequences [30]. This seems to be an unspoken truth. Too many of the surgical randomized controlled trials also lack a focus on specific surgical details [31]. Generalizability is an additional problem, as it remains difficult to extrapolate—even valid—RCT results to the real world. To apply research results into clinical practice, a fundamental contribution of the clinician is needed: their personal experience and the technical and organizational situation of their working environment must be adequate for the translation process. This might require important changes that require time, money, and administrative changes. This is probably the main reason why high-level evidence is lacking for essential surgical conditions. The recommendations of the AAOS for surgical interventions are often based on consensus, and often, very strong evidence for important decisions is missing or left open. The questions raised are why more robust research is still needed and what the obstacles to formulating reliable guidelines are [32]. The goal must be to find measures that could improve the situation in surgery and make it resemble that of internal medicine.

### 2.2. Data Formats

At present, data formats like those proposed by the ICUC working group [8,22,33,34,35] are not routinely produced and included in scientific publications. How technical performance influences outcomes is therefore left to speculation or deduced from secondary data. Instead, the presentation of data should allow a direct comparison with golden standard data, hopefully defined not only verbally but with images. [18]. According to Ioannidis [31], “the production of systematic reviews and meta-analyses has reached epidemic proportions” and unfortunately often presents “redundant, misleading and conflicted” data. Zavalis et al. [23] correctly asks for more comparisons between drug therapy and surgery. More comparisons should also be available between surgery and conservative treatment without drugs. This is only possible if almost-matched pairs can be directly compared in accessible complete datasets.

The construction of a comprehensive image dataset must adhere to the Health Insurance Portability and Accountability Act (HIPAA) standards for de-identification protocols and necessitates patient consent being obtained [36]. With technological advancements, the feasibility of accessing massive still images and short video clips on the cloud has increased, and the associated costs have decreased [37]. For example, the ICUC registry database consistently accumulates prospective orthopedic trauma cases in accordance with the specified guidelines [34].

### 2.3. Publication Formats

The advancements in information technology should now allow adapted publication formats to present the high data volume resulting from a complete documentation format by a link. Furthermore, it seems legitimate to assume that, in the near future, supervised machine learning and pattern recognition will facilitate the analysis of big image data. This would also allow us to better understand the details of surgical procedures. As a result, current paradigms must be challenged by the collaboration of scientists, surgeons, industry, and regulatory authorities. The clinical expertise of the individual surgeon and their intra-operative performance quality represent the most relevant components of Evidence-Based Surgery [25,30].

Along with the careful consideration of indications, clinical expertise includes the careful indication and a sufficient number of treated cases for a defined clinical problem. The correlation between the caseload and outcome quality is sound. However, the precise number needed to reach the expertise level is difficult to define and can vary from one surgeon to another [38].

The transition to digital technology indicates that information is increasingly being sourced from alternative sources to textbooks, such as mobile phones. Yet, the data used for “guidelines” still suffer from the abovementioned “technical performance bias [39].” The results are studies that suffer from a “lack of homogeneity of patient series” [22,31]. Having RCTs with complete datasets that are amenable to secondary analysis would be a major step forward in bringing surgery to the level of medical studies. Furthermore, having complete datasets accessible to peers for the reviewing process would improve the selection of relevant publications. As previously mentioned, a clinical trial unit can provide significant methodological help to surgeons to produce valid RCTs [40]. The same is possible thanks to institutions like the National Institute for Health and Care Research—Complex Reviews Support Unit (NIHR CRSU) [41].

## 3. Further Obstacles to Evidence-Based Medicine in Surgery

Several factors lead to the challenges in conducting surgical RCTs. Among the surgeon-related factors, a lack of corresponding basic training and knowledge of epidemiology and statistics might be relevant. This can easily be overcome by clinical trial units (CTUs), i.e., specialized biomedical research units that coordinate and help clinicians to design and analyze clinical trials and other studies [40]. The advantages of such units are undisputed all over the world. A second factor limiting the number of useful RCTs related to surgery concerns surgical techniques. Surgical procedures are composed of more individual steps than medical treatments. Thus, analyzing the single elements and defining a correct, stepwise protocol is more complex.

### 3.1. Steps and Details of a Surgical Procedure

Standardizing a surgical procedure’s key steps is difficult when different surgeons or even multiple centers are involved. Rapid changes in technology, research, or institutional organization can render protocols rapidly obsolete before the end of a study. Even with the most accurate wording, including schematic drawings or images of single clinical examples, execution differences from one surgeon to another cannot be avoided. This “technical performance bias” [35] can only be recognized when complete intra-operative image documentation is available and secondary analysis of the whole series is allowed [8]. For instance, Lans et al. presented a study using a radiolucent clamp to reduce the distal radius’s dorsal lunate facet fracture [42]. By providing comprehensive surgical images, including radiographs and computed tomography, we enable independent reviewers or readers to critically analyze the surgical outcomes. Surgeons confronted with a comprehensive dataset to find similar or matched cases and might find useful technical hints or alternatives. 

Unfortunately, such documentation is rarely available. Numerous studies display poor reporting and thus are at a high risk of bias [43]. The surgical world is, therefore, confronted with what can be called a crisis of reproducibility [44].

### 3.2. Documentation Tools

Specialties that utilize endoscopy techniques have begun to compile comprehensive intra-operative documentation by recording the full video streams of the procedures. This practice has underscored the critical role that the quality of technical performance plays in determining the success of a surgical treatment [24,45]. The workup of lengthy video streams is possible, but it is quite cumbersome and needs time-consuming tagging if performed secondarily. If performed intra-operatively, it might increase the operation time or distract the surgeon. A complete sequence of still images and short video clips of the key steps would be easier. 

Comprehensive documentation of surgical interventions enhances the development of supervised machine learning algorithms. Advanced artificial intelligence (AI) technologies facilitate the automated post-processing of crucial still images and concise short video clips, ensuring rapid access and precision. AI-integrated surgical platforms have played a pivotal role in select endoscopic-assisted surgeries [46], indicating their potential utility across various surgical techniques, such as arthroscopic surgeries, in future clinical settings.

### 3.3. Secondary Data Analysis, Comparisons, and Compliance with the Protocol

Given that surgical protocols are pivotal elements of a study, without a possible secondary analysis of the abovementioned intra-operative performance documented by images, it will be exceedingly difficult to know how precisely a written protocol has been followed. Confounding variables such as different centers, surgical skills, and tacit knowledge between different surgeons will also influence the technical performance. It remains surprising that there is not more pressure to ask for complete intra-operative image documentation for surgical publications [47]. Surgeons and publishers seem to like time-honored concepts and teachings [48] and are, therefore, slow to reject them.

A significant issue is the sheer volume of guidelines and algorithms available in the current landscape. Professionals, regardless of their age, find it increasingly challenging to navigate this abundance while adhering to tight schedules and striving to maintain a healthier work–life balance compared to their more senior colleagues.

### 3.4. Other Obstacles

Certain obstacles to Evidence-Based Surgery will remain, not the least of which is that double blinding will remain difficult, as well as considering the inclusion of placebo surgery [49,50]. Formal scientific investigations will never replace what one can learn from years of personal or teaching experience, especially if this experience is continuously questioned [51]. In a modern world, continuous learning [52] is mandatory for everybody. Part of the learning content—especially for surgical trainees—lies outside the world of EBM and explicit knowledge. The essential field of surgical decision-making is difficult to teach. Surgeons must be decisive, even when uncertain and often under emergencies when lengthy consultations are impossible. They then rely on truthiness, i.e., concepts that are believed to be true, regardless of information like facts and analysis, which might provide contradictory information [21,53].

## 4. Wish for the Future: Pathways to Sound Evidence-Based Surgery

### 4.1. Improved Guidelines, Shared Decision-Making, and Complete Data in Surgical Trials

Surgeons would appreciate to ability to base their daily decisions on evidence. Recognizing this, the American College of Surgeons produced a collection of electronic guideline books, “Evidence-Based Decisions in Surgery” [54]. The available guidelines are, or at least can be, an important help for surgeons. They are the product of lengthy discussions between renowned specialists. However, when confronted with a given case, it might be that no guideline fits, because the case is rare or does not correspond to a special type of classification used in the guidelines. Increasingly often, such cases are discussed during regular multidisciplinary team meetings, a well-established practice in oncology. Guidelines are based on evidence but do not consider the goals, preferences, and special situations of the patients.

If an emergency procedure is needed in orthopedic trauma, time can be gained by following the concept of DCO, damage control orthopedics [55], which partially replaces ETC, early total care. In orthopedic trauma, the use of an external fixator and the postponement of definite surgery is proposed, leaving time for input from other colleagues.

For elective cases, an additional trend is SDM, shared decision-making. Its adoption might be slowed by concerns about the potential legal consequences of its use [56]. Nevertheless, it should become routine in the future. SDM is not a new paradigm, as it is an essential part of Evidence-Based Medicine [57]. Surgical patients prefer SDM as they prefer less or non-invasive treatment options. Future studies should evaluate whether SDM lowers the incidence of the overuse of surgery. However, SDM should also include patient preferences. This element is not yet sufficiently implemented, despite being an ethical and legal obligation. The importance of the patient’s involvement appears when we realize that there might be relevant differences in the outcome measures that the surgeon considers decisive and those of the patients. Surgeons might prefer to prolong life and improve function at almost any cost, whereas patients might prefer to have fewer surgical interventions, shorter hospital stays, and shorter pain periods. Balancing these two views is difficult, and the final decision must be made after the patient has been given comprehensive and honest information. The principle “voluntas aegroti suprema lex,” which translates to “the will of the patient is the supreme law,” has supplanted the earlier guideline “salus aegroti suprema lex,” meaning “the well-being of the patient is the supreme law.” This shift underscores a modern ethical framework that prioritizes patient autonomy and individual choice over paternalistic concerns for the patient’s well-being. They, therefore, could consider this to be a better quality of life, even at the cost of a shorter survival period. This has had an important impact on outcome measurements, making PROMs, patient-related outcome measurements, today’s preferred modern method [58].

Surgeons might have preferences that can even be evidence-based, but mostly, there are alternatives, which might have equivalent evidence bases. The patient must be informed accordingly to respect their autonomy. If the alternative does not coincide with the surgeon’s expertise, the doctor should propose a colleague. SDM helps in such a situation, as the patient must know from the beginning that alternatives exist, and a choice will be needed. If confronted with uncertainty, a neutral information fashion is even more important.

### 4.2. Intra-Operative Image Data Recording

Comparable and reproducible data retrieval is an essential component of evidence production. The frequency of data recording in the OR is increasing, but guidelines do not yet exist, as surgical data science is a new field [59]. The purposes of the data collection are very different, as are the technical modalities for their realization. A clear definition of the purpose helps to avoid the collection of data that are not used secondarily. A broad agreement on the details of data collection among different centers would improve the quality of primary data. Asking for complete intra-operative image data from case series in surgical RCTs would be an important contribution to reproducibility and comparison problems [8]. Achieving reproducibility is essential for the credibility of research data. 

### 4.3. Avoid Inconclusive Reports

Inconclusive results in RCTs comparing different techniques or different results from different trials may result from having inhomogeneous groups. Missing details in the data presentation make any comparison impossible. Even the best-written protocols for a surgical procedure bear the risk of ambiguity and interpretation problems, even when drawings are added. Retrospectively, it is difficult to judge whether the protocol has been homogeneously followed. Therefore, by adding complete data, the scientific value of RCTs is significantly increased. It can help surgery to fill the gap to internal medicine.

Improved intervention descriptions in surgical trials have been recognized to be necessary for almost two decades [60], but astonishingly, an explicit request for complete intra-operative image documentation is rare [59]. Complete intra-operative image data also allow access to intra-operative problems and harm, which are often not documented and reported with maximal transparency. Inadequate reporting was found in 33 of 88 studies by Stoubenrouch et al. [1]. Such data are also valuable learning instruments. Not only should harmful intra-operative actions be documented, but also, suboptimal actions that can easily be improved (e.g., the use of special instruments or skilled reduction maneuvers in fracture treatment or less invasive access techniques) should be recorded [61]. Modern information technology allows the presentation of such details in a link, thus not excessively lengthening the main paper. In addition, this allows every user to pick what they need. Complete data also include details about the use of special technology, like the number of fluoroscopy shots and time, as they reflect observations of the as low as reasonably achievable (ALARA) principles [62]. There is a difference if the same (perfect) reduction is obtained with 20 or 100 fluoroscopic shots. Such data are rarely reported [33]. This is particularly important given the growing and scientifically substantiated trend of utilizing three-dimensional imaging techniques during surgery [63].

Future studies in orthopedics should explicitly address the concerns associated with the incomplete reporting of performance measures and the inadequate handling of missing data [43,64]. Surgeons from any curriculum level like case discussions [65]. They allow them to utilize the knowledge acquired from textbooks and literature in real-world situations. The more complex the cases are, the more the singularity of every case appears. Most frequently, there is more than one valid solution. This is not only true for the technical tools used, but also for the concepts applied. This is best illustrated by using almost matched pairs with results for different concepts or tools. The singularity of every case appears even more clearly for rare pathologies and complications, which are often the most challenging cases. Classifications have been developed to overcome the difficulties and define common rules. However, the frequent lack of inter-observer reliability reduced the attempts.

### 4.4. Joint Efforts

Modern high-quality research is very expensive. Funding bodies are indispensable, but conflicts of interest must be avoided by clear rules to guarantee complete transparency. Outcome studies should focus on factors relevant to patients (PROMs), and the outcome data should be provided by surgeons who are not directly involved in the surgery. Functional and pain level data should be obtained during every surgeon contact and not only at the end of the study. Protocols that allow telemedical data production should be preferred. Patients might be incentivized to participate in trials by incentives and people independent from the treatment team, but the risk of biases must be carefully considered. The CONSORT perfectly addresses ethical issues.

The US government will guarantee open access to all research publications it funds. The policy will go into effect in 2026 and apply to everything that receives federal money. Similar decisions have been made in other countries, for example, the Swiss National Science Foundation in Switzerland.

The “globalized” world with many different healthcare systems exposes impressive differences concerning access to facilities and technologies. Future “guidelines” should also consider “global real-world data” and not only data resulting from the special situations considered in randomized controlled trial protocols. By such measures, surgical decision-making for surgeons would become more evidence-based and no longer biased by tradition “on what works in my hands and/or what my mentor told me” [66,67]. 

To improve evidence in surgery, a joint effort is needed to guarantee good organizational support to ensure that adequate personnel, resources, and funding exist for those surgeons willing to improve Evidence-Based Surgery. More efforts are needed to educate young surgeons to define relevant clinical problems and analyze whether different techniques can solve the problems associated with a given surgery. “Shared decision-making” might also help with the management of complex clinical information combined with structured patient information [48,68].

### 4.5. New Publication Formats

New publishing formats [8,69,70] with shorter text precisely defining a clinical problem, showing examples of different valid solutions and allowing access to complete data through links would make scientific reading more efficient by avoiding redundant descriptions of basic knowledge. Scrolling through many cases in such a link would be an excellent help for surgeons who have to treat a specific clinical problem and are searching for similar cases [35]. For example, it is practical for a modern publication to embed a link to access the database containing all relevant surgical images [12].

### 4.6. Documentation Tools

Technical progress will soon offer relatively simple, affordable head-mounted devices for the image documentation of daily surgical practice [71]. Up to now, these devices have mainly been used for augmented reality use in urology and neurosurgery in university settings. More and more, such devices will not only facilitate global learning and telemedical communication but will also help to fill the gap between countries with different economic resources. This seems necessary, considering that an important percentage of surgical procedures are performed in regions with limited resources and sometimes completely lacking documentation [72].

It will be interesting to see whether the commercial evolution and data protection legislation will allow these tools to enter the medical arena, i.e., the operation room, and produce relevant help to surgeons globally.

## 5. Conclusions

Complete intra-operative imaging can offer readers and reviewers an assessment of the technical performance quality. Such an approach can also be integrated into RCTs, potentially minimizing the variability within surgical treatment groups. More stringent reporting should be encouraged by researchers, authors, and especially journal editors. The data should already be accessible to the experts for peer-reviewing. The detailed data should be separated from the main text and should be accessible by a link, creating a new lean publication format. The main text should be as short as possible to avoid redundant information. This allows surgeons to gain a more comprehensive understanding when interpreting the results of clinical trials or observational studies, which can be beneficial for advancing Evidence-Based Surgery.

## Data Availability

Not applicable.

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
