# Peer review of "Evidence-Based Surgery: What Can Intra-Operative Images Contribute?"

_jcm, 2023, doi:10.3390/jcm12216809_

Round 1

Reviewer 1 Report

Comments and Suggestions for Authors

This manuscript presents a sophisticated exploration of the significance of comprehensive intra-operative imagery in advancing evidence-centric surgical procedures. The authors construct a robust case illustrating the deficiencies in surgical randomized controlled trials regarding intervention reporting. This shortfall obstructs the precise assessment of technical competence and impairs reproducibility. The manuscript advocates for access to holistic intra-operative image archives, asserting that such accessibility would permit an auxiliary analysis of procedure quality and expertise, thus paving the way for pertinent comparisons among surgical professionals and across trials.

Several primary advantages of exhaustive image data are delineated:

• Permits a thorough evaluation to confirm uniform adherence to surgical protocols, thereby minimizing confounding factors linked to technical prowess. This could potentially mitigate ambiguous or conflicting outcomes rooted in non-uniform patient cohorts.

• Acts as an invaluable pedagogical tool, allowing stakeholders to scrutinize comprehensive data, pinpointing elements most pertinent to their clinical practice.

• Supports the pragmatic translation of research conclusions into tangible clinical implementations by establishing a benchmark for technical expertise.

• Promotes transparency concerning intra-operative complications and adverse events.

To realize this ambitious paradigm, the authors methodically outline actionable measures encompassing the formulation of protocols for systematic intra-operative data aggregation, adoption of head-mounted camera systems, introduction of novel publication blueprints linked to all-inclusive datasets, and harnessing AI and machine learning capacities for vast dataset analyses. They astutely emphasize the imperative of collaborative endeavours spanning multiple stakeholders.

This viewpoint aligns congruently with the escalating advocacy for transparent, replicable research. The authors contend that such initiatives could potentially elevate surgical research standards, aligning them with those observed in internal medicine studies.

Refinements for peer review consideration:

• Delve deeper into the tangible advantages and applications conferred by comprehensive intra-operative datasets. Illustrate with one or two precedent studies where such data might have elucidated findings or permitted superior analyses.

• Concisely address conceivable challenges from mandatory comprehensive image datasets—cost implications, patient confidentiality, and data storage requisites. Propose mitigating strategies.

• Offer explicit details concerning the envisioned innovative publication designs, elucidating how these could present datasets accessibly without unduly extending article lengths.

• Maintain a constructive, affirmative tone. Refrain from overtly critical rhetoric regarding existing research lacunae. Highlight the prospective merits of the recommended strategy.

• For enhanced validity, contemplate embedding an external perspective in alignment with the authors’ standpoint, such as referencing a source advocating analogous enhancements in surgical trial documentation.

• Guarantee that the review consistently emphasizes the manuscript’s pivotal theme: the imperative for comprehensive intra-operative image data. Ancillary subjects should be brief or refined if they deviate from this principal argument.

• Recommend any pertinent academic works not currently cited that could bolster the authors’ propositions, ensuring the manuscript is positioned within the broader scholarly conversation on the subject.

Minor modifications:

• Cite specific instances where complete intra-operative data might have enriched or clarified outcomes in previous surgical investigations to underscore potential ramifications.

• Provide a more detailed exposition of the proposed innovative publication modalities tailored to present exhaustive datasets.

• Reflect upon potential constraints or drawbacks of mandating comprehensive intra-operative image data, such as cost implications or patient privacy issues.

In conclusion, this manuscript is an astute, eloquently articulated discourse on a pivotal subject. The authors adeptly amalgamate diverse evidence into pragmatic recommendations to enhance surgical research paradigms substantially. Their insights are commendable.

Comments on the Quality of English Language

Minor editing is required.

Author Response

This manuscript presents a sophisticated exploration of the significance of comprehensive intra-operative imagery in advancing evidence-centric surgical procedures. The authors construct a robust case illustrating the deficiencies in surgical randomized controlled trials regarding intervention reporting. This shortfall obstructs the precise assessment of technical competence and impairs reproducibility. The manuscript advocates for access to holistic intra-operative image archives, asserting that such accessibility would permit an auxiliary analysis of procedure quality and expertise, thus paving the way for pertinent comparisons among surgical professionals and across trials.

Several primary advantages of exhaustive image data are delineated:

  • Permits a thorough evaluation to confirm uniform adherence to surgical protocols, thereby minimizing confounding factors linked to technical prowess. This could potentially mitigate ambiguous or conflicting outcomes rooted in non-uniform patient cohorts.
  • Acts as an invaluable pedagogical tool, allowing stakeholders to scrutinize comprehensive data, pinpointing elements most pertinent to their clinical practice.
  • Supports the pragmatic translation of research conclusions into tangible clinical implementations by establishing a benchmark for technical expertise.
  • Promotes transparency concerning intra-operative complications and adverse events.

To realize this ambitious paradigm, the authors methodically outline actionable measures encompassing the formulation of protocols for systematic intra-operative data aggregation, adoption of head-mounted camera systems, introduction of novel publication blueprints linked to all-inclusive datasets, and harnessing AI and machine learning capacities for vast dataset analyses. They astutely emphasize the imperative of collaborative endeavours spanning multiple stakeholders.

This viewpoint aligns congruently with the escalating advocacy for transparent, replicable research. The authors contend that such initiatives could potentially elevate surgical research standards, aligning them with those observed in internal medicine studies.

Refinements for peer review consideration:

  1. Delve deeper into the tangible advantages and applications conferred by comprehensive intra-operative datasets. Illustrate with one or two precedent studies where such data might have elucidated findings or permitted superior analyses.

Ans: Thank you for your great comments. We present two cases, one involving a common fracture and the other a relatively rare fracture, utilizing complete intra-operative images derived from our database for secondary analysis in Lines 80-86.

For example, in a secondary workup of a surgical database containing comprehensive intra-operative imagery, we can analyze prevalent fractures such as the joint depression type of tibial plateau fractures [12]. Furthermore, we can also evaluate less common fractures like dorsal radiocarpal dislocations [13]. In this paper, we focus on the importance of intra-operative image data and do not expand on statistical, epidemiological, and outcome aspects. The importance of corresponding aspects is well-defined in the literature [10,14].

  1. Concisely address conceivable challenges from mandatory comprehensive image datasets—cost implications, patient confidentiality, and data storage requisites. Propose mitigating strategies.

Ans: Thank you for your comments. We added a paragraph in the Data Format to provide an example that meets the criteria of patient confidentiality and technological accessibility in Lines 155-161.

The construction of a comprehensive image dataset must adhere to the Health Insurance Portability and Accountability Act (HIPAA) standards for de-identification protocols and necessitates obtaining patient consent [36]. With technological advancements, the feasibility of accessing massive still images and short video clips on the cloud has increased, and the associated costs have decreased [37]. For example, the ICUC registry database consistently accumulates prospective orthopedic trauma cases in accordance with the specified guidelines [34].

  1. Offer explicit details concerning the envisioned innovative publication designs, elucidating how these could present datasets accessibly without unduly extending article lengths.

Ans: Thank you for your great comments. We demonstrated a new publication format and design with an embedded link to access the database containing complete images in section 4 New Publication Format, Lines 383-389.

New publishing formats [8,70,71] with a shorter text precisely defining a clinical problem, showing examples of different valid solutions, and allowing access to complete data through links would make scientific reading more efficient by avoiding redundant descriptions of basic knowledge. Scrolling through many cases in such a link would be an excellent help for surgeons who have to treat a specific clinical problem and are searching for similar cases [35]. For example, it is practical for a modern publication to embed a link to access the database containing all relevant surgical images [12].

  1. Maintain a constructive, affirmative tone. Refrain from overtly critical rhetoric regarding existing research lacunae. Highlight the prospective merits of the recommended strategy.

Ans: Thank you for your comments. This article is not intended to critique the role of RCTs in surgery within the context of evidence-based surgery. Instead, we aim to highlight the challenges and limitations of implementing RCTs in surgery. We suggest that complete intra-operative imaging can offer readers and reviewers an assessment of surgical performance. Such an approach can also be integrated into RCTs, potentially minimizing the variability within surgical treatment groups. This allows surgeons a more comprehensive understanding when interpreting the results of clinical trials, which can be beneficial for advancing evidence-based surgery. We rewrote the Conclusions in Lines 403-412.

Complete intra-operative imaging can offer readers and reviewers an assessment of the technical performance quality. Such an approach can also be integrated into RCTs, potentially minimizing the variability within surgical treatment groups. More stringent reporting should be encouraged by researchers, authors, and especially journal editors. The data should already be accessible to the experts for peer-reviewing. The detailed data should be separated from the main text and accessible by a link, creating a new, lean publication format. The main text should be as short as possible to avoid redundant information. This allows surgeons a more comprehensive understanding when interpreting the results of clinical trials or observational studies, which can be beneficial for advancing evidence-based surgery. 

  1. For enhanced validity, contemplate embedding an external perspective in alignment with the authors’ standpoint, such as referencing a source advocating analogous enhancements in surgical trial documentation.

Ans: Thank you for your comment. We revised the paragraph and tried to mention the importance of surgical performance and surgical documentation with medical images in Lines 105-114.

The difficulty in creating homogeneous groups in developing surgical randomized controlled trials has been identified as an important problem [22]. In medical and surgical trials, it is possible to assign patients to different treatment arms. However, the evaluation of the medical outcome of the administration of a pharmaceutical may prove clearer than that of a surgical procedure [23]. One important reason could be the fact that contemporary documentation formats rarely allow the direct assessment of differences in surgical skills, which have a direct impact on the outcomes [24,25]. A visual representation conveys complex data more effectively than a textual description in surgical documentation [26]. Moreover, a complete intraoperative image documentation will allow us to analyze the skill-dependent differences in technical performance [22].

  1. Guarantee that the review consistently emphasizes the manuscript’s pivotal theme: the imperative for comprehensive intra-operative image data. Ancillary subjects should be brief or refined if they deviate from this principal argument.

Ans: Thank you for your feedback. We have removed certain relatively unrelated ancillary subjects, such as the section on the Potential of Modern Technology.

  1. Recommend any pertinent academic works not currently cited that could bolster the authors’ propositions, ensuring the manuscript is positioned within the broader scholarly conversation on the subject.

Ans: Thank you for your suggestions. To comprehensively discuss evidence-based surgery, more literature is still needed, as mentioned in Lines 49-51.

Such a project is unsuitable for a single article; a whole book is needed, like the one edited by Thoma et al. a few years ago [3].

A dedicated monograph might be required to elucidate this topic thoroughly. Nonetheless, this article focused on some challenges faced in evidence-based surgery, such as inadequate assessment of surgical performance. We present our perspectives, practical implementations of surgical databases, and published literature. With 73 references cited, this evidence can persuade surgeons to accept our proposed views.

Minor modifications:

  • Cite specific instances where complete intra-operative data might have enriched or clarified outcomes in previous surgical investigations to underscore potential ramifications.

Ans: Thank you for your great comments. We present two cases, one involving a common fracture and the other a relatively rare fracture, utilizing complete intra-operative images derived from a surgical database with complete intra-operative images for secondary analysis.
The instances were mentioned in Lines 80-86 and Lines 205-209.

For example, in a secondary workup of a surgical database containing comprehensive intra-operative imagery, we can analyze prevalent fractures such as the joint depression type of tibial plateau fractures [12]. Furthermore, we can also evaluate less common fractures like dorsal radiocarpal dislocations [13]. In this paper, we focus on the importance of intra-operative image data and do not expand on statistical, epidemiological, and outcome aspects. The importance of corresponding aspects is well-defined in the literature [10,14].

For instance, Lans et al. presented a study using a radiolucent clamp to reduce the distal radius's dorsal lunate facet fracture [42]. By providing comprehensive surgical images, including radiographs and computed tomography, we enable independent reviewers or readers to critically analyze the surgical outcomes.

  • Provide a more detailed exposition of the proposed innovative publication modalities tailored to present exhaustive datasets.

Ans: Thank you for your great comments. We demonstrated a new publication format and design with an embedded link to access the database containing complete images in Lines 383-389.

New publishing formats [8,70,71] with a shorter text precisely defining a clinical problem, showing examples of different valid solutions, and allowing access to complete data through links would make scientific reading more efficient by avoiding redundant descriptions of basic knowledge. Scrolling through many cases in such a link would be an excellent help for surgeons who have to treat a specific clinical problem and are searching for similar cases [35]. For example, it is practical for a modern publication to embed a link to access the database containing all relevant surgical images [12].

  • Reflect upon potential constraints or drawbacks of mandating comprehensive intra-operative image data, such as cost implications or patient privacy issues.

Ans: Thank you for your comments. We added a paragraph in Lines 155-161 to demonstrate a real-world example with a complete surgical image database adhering to patient privacy.

The construction of a comprehensive image dataset must adhere to the Health Insurance Portability and Accountability Act (HIPAA) standards for de-identification protocols and necessitates obtaining patient consent [36]. With technological advancements, the feasibility of accessing massive still images and short video clips on the cloud has increased, and the associated costs have decreased [37]. For example, the ICUC registry database consistently accumulates prospective orthopedic trauma cases in accordance with the specified guidelines [34].

  • In conclusion, this manuscript is an astute, eloquently articulated discourse on a pivotal subject. The authors adeptly amalgamate diverse evidence into pragmatic recommendations to enhance surgical research paradigms substantially. Their insights are commendable.

Ans: Thank you for your comments.

Reviewer 2 Report

Comments and Suggestions for Authors

Authors have compiled one of the crucial topics in Evidence-Based Surgery. Although the manuscript is reasonably good, to improve the quality of the manuscript the following review suggestions were provided below.

1)     Abstract seems to be vague and lacks in coherency between the previous sentences. The current state of art should be clearly conveyed to delineate the prime novelty of this review.

2)     Introduction is too small, first of all authors should provide a basic panoramic overview on what is evidence-based surgery. Its series of developments till date? Its applications with respect to diverse domains (ortho, gastro, cardio and so on). What is the recent survey statics indices on evidenced based surgery from the viewpoint of Intra-operative Images? It should also clearly convey what are the current draw backs hampering the development of evidence-based surgery.

3)     Section 3 and 4 were written only in surface level. It would be nice if authors could expand the number of literatures and sum up the data. For instance, it is good to include lots of case studies data and highlight the how intra operative images were significant in that study. Section 3.2 and 3.3 only very few techniques were presented. Still there are lot many to highlight which would interest broad sets of audience and attract citations.

4)     The percentage of improvement that evidence based surgery can bring is nowhere highlighted.

Author Response

Authors have compiled one of the crucial topics in Evidence-Based Surgery. Although the manuscript is reasonably good, to improve the quality of the manuscript the following review suggestions were provided below.

  • Abstract seems to be vague and lacks in coherency between the previous sentences. The current state of art should be clearly conveyed to delineate the prime novelty of this review.

Ans: Thank you for your comments. We rewrote the abstract according to your suggestions in Lines 15-36.

Evidence-based medicine integrates results from randomized controlled trials (RCTs) and meta-analyses, combining the best external evidence with individual clinical expertise and patients' preferences. However, many RCTs of surgery differ from medicine in that surgical performance is often assumed to be consistent. Yet, evaluating whether each surgery is performed to the same standard is quite challenging. As a primary issue, the novelty of this review is to emphasize – with a focus on orthopedic trauma - the advantage of complete intra-operative image documentation, allowing the direct evaluation of the quality of the intraoperative technical performance. The absence of complete intraoperative image documentation leads to inhomogeneity of case series, yielding inconsistent results due to the impossibility of secondary analysis. Thus, comparisons and reproduction of studies are difficult. Access to complete intra-operative image data in surgical RCTs not only allows secondary analysis but also comparison to similar cases. Such complete data can be included in electronic papers. Offered to peers – in an accessible link - when presenting papers facilitates the selection process and improves publications for readers. Additionally, having access to the full set of image data, for all the presented cases serves as a rich resource for learning. It enables the reader to sift through the information and pinpoint the details that are most relevant to his individual needs, potentially incorporating this knowledge into daily practice. A broad use of the concept of complete intraoperative image documentation would be pivotal in bridging the gap between clinical research findings and real-world applications. Enhancing the quality of surgical RCTs would facilitate the equalization of evidence acquisition in both internal medicine and surgery. A joint effort of surgeons, scientific societies, publishers, and healthcare authorities is needed to support the ideas, implement economic requirements, and overcome the mentality obstacles to its realization.

  • Introduction is too small, first of all authors should provide a basic panoramic overview on what is evidence-based surgery. Its series of developments till date? Its applications with respect to diverse domains (ortho, gastro, cardio and so on). What is the recent survey statics indices on evidenced-based surgery from the viewpoint of Intra-operative Images? It should also clearly convey what are the current drawbacks hampering the development of evidence-based surgery.

Ans: Thank you for your comments. We expanded the introduction in Lines 59-65 and 74-86.

The core principles of evidence-based medicine are established through many randomized controlled trials (RCTs) and meta-analyses. However, heterogeneity in surgical trials, such as surgeon-related factors (including surgeon preferences, factors (such as reluctance to participate and patient preferences), and methodological challenges (inclusion/exclusion criteria, sample size estimation, follow-up/drop-out rate, and funding can make it difficult to conduct RCTs in surgery [4-6]. In certain circumstances, observational cohort studies may provide insights comparable to those of RCTs [7].

This paper, therefore, starts from the actual situation of evidence-based orthopedic surgery in a broad sense and focuses on the mentioned importance of intra-operative image data and the homogeneity of case groups. With our experience of data- set of more than 1000 orthopedic trauma cases, with complete intra-operative image documentation including radiologic imaging and corresponding follow-up [11], we think we have valid arguments for the usefulness of such data. Orthopedic trauma is only a small segment of orthopedic surgery, but we feel comfortable drawing more extensive conclusions. For example, in a secondary workup of a surgical database containing comprehensive intra-operative imagery, we can analyze prevalent fractures such as the joint depression type of tibial plateau fractures [12]. Furthermore, we can also evaluate less common fractures like dorsal radiocarpal dislocations [13]. In this paper, we focus on the importance of intra-operative image data and do not expand on statistical, epidemiological, and outcome aspects. The importance of corresponding aspects is well-defined in the literature [10,14].

  • Section 3 and 4 were written only in surface level. It would be nice if authors could expand the number of literature and sum up the data. For instance, it is good to include lots of case studies data and highlight the how intra operative images were significant in that study. Section 3.2 and 3.3 only very few techniques were presented. Still there are lot many to highlight which would interest broad sets of audience and attract citations.

Ans: Thank you for the comments. We expanded section 3 and mentioned an example of a current AI application for retrieving still images or short video clips from the surgical recording in Lines 223-228.

Comprehensive documentation of surgical interventions enhances the development of supervised machine-learning algorithms. Advanced artificial intelligence (AI) technologies facilitate automated post-processing of crucial still images and concise short video clips, ensuring rapid access and precision. AI-integrated surgical platforms have played a pivotal role in select endoscopic-assisted surgeries [46], indicating potential utility across various surgical techniques, such as arthroscopic surgeries, in future clinical settings.

We also provided an example of how intraoperative images provide significance in evaluating surgical performance in Lines 205-209.

For instance, Lans et al. presented a study using a radiolucent clamp to reduce the distal radius's dorsal lunate facet fracture [42]. By providing comprehensive surgical images, including radiographs and computed tomography, we enable independent reviewers or readers to critically analyze the surgical outcomes.

  • The percentage of improvement that evidence-based surgery can bring is nowhere highlighted.

Ans: Thanks for your comments. To improve evidence in surgery, a joint effort is needed to guarantee good organizational support to ensure adequate personnel, resources, and funding exist for those surgeons willing to improve Evidence-Based Surgery. More efforts are needed to educate young surgeons to define relevant clinical problems and analyze whether different techniques can solve the problems of a given surgery. “Shared decision-making” might also help to manage complex clinical information combined with structured patient information. We highlighted what improvement of evidence-based surgery can bring in Lines 375-381.

To improve evidence in surgery, a joint effort is needed to guarantee good organizational support to ensure adequate personnel, resources, and funding exist for those surgeons willing to improve Evidence-Based Surgery. More efforts are needed to educate young surgeons to define relevant clinical problems and analyze whether different techniques can solve the problems of a given surgery. “Shared decision-making” might also help to manage complex clinical information combined with structured patient information [48,69].
